# The Potential Role of Renal Denervation in the Management of Heart Failure

**DOI:** 10.3390/jcm11144147

**Published:** 2022-07-17

**Authors:** Kameel Kassab, Ronak Soni, Adnan Kassier, Tim A. Fischell

**Affiliations:** 1Division of Cardiology, Borgess Heart Institute, 1521 Gull Road, Kalamazoo, MI 49048, USA; ronak.soni@ascension.org (R.S.); adnan.kassier@ascension.org (A.K.); tafisc@gmail.com (T.A.F.); 2Division of Cardiology, Michigan State University, Kalamazoo, MI 49048, USA

**Keywords:** renal denervation, heart failure, therapies

## Abstract

Sympathetic nervous system activation in patients with heart failure is one of the main pathophysiologic mechanisms associated with the worse outcomes. Pharmacotherapies targeting neurohormonal activation have been at the center of heart failure management. Despite the advancement of therapies and the available treatments, heart failure continues to have an overall poor prognosis. Renal denervation was originally developed to lower systemic blood pressure in patients with poorly controlled hypertension, by modulating sympathetic outflow. However, more recently, multiple studies have investigated the effect of renal denervation in heart failure patients with both preserved (HFpEF) and reduced ejection fractions (HFrEF). This paper provides an overview of the potential effect of renal denervation in altering the various pathophysiologic, sympathetically mediated pathways that contribute to heart failure, and reviews the literature that supports its future use in those patients.

## 1. Introduction

Heart failure remains among the leading causes of morbidity and mortality in the United States, with a continuously increasing prevalence of the disease [1]. Despite the advancements in heart failure therapies, from pharmacotherapies to mechanical support devices and transplant, the morbidity and mortality have remained essentially unchanged [2]. Heart failure is a complex syndrome, characterized by cardiac remodeling under the effect of multiple pathophysiological mechanisms. In chronic cardiovascular diseases, including heart failure, the sympathetic nervous system is constantly being activated. Initially, this activation is compensatory and necessary for the regulation of cardiac output and systemic perfusion. However, chronically, it becomes maladaptive, leading to unfavorable cardiac and vascular remodeling [3,4]. Renal sympathetic activation, and cross-talk with the central nervous system (nucleus tractus solitarius), is one of the regulatory pathways of cardiac parameters, such as heart rate, blood pressure, and cardiac output. The feedback pathway between the kidneys and the CNS is one of the major regulators of central sympathetic outflow. They produce neuro-hormonal modulation via the activation of the renin–angiotensin–aldosterone system (RAAS) in response to hypoperfusion, leading to sodium retention, vasoconstriction, and intravascular volume depletion [5]. A regional norepinephrine (NE) spillover in the kidneys and the heart has been correlated with a worse prognosis compared to systemic norepinephrine levels, which may make renal sympathetic pathways a reasonable therapeutic target [6]. In this review, we evaluate the pathophysiological mechanisms linking sympathetic activation to heart failure pathogenesis, and evaluate the evidence supporting a potential use of renal denervation as a therapeutic option for patients with heart failure, with both reduced and preserved ejection fraction.

## 2. Sympathetic Activation in Heart Failure

In chronic heart failure, the complex interplay between the neuro-hormonal, sympathetic, and inflammatory pathways provides the pathophysiologic milieu for cardiac remodeling [7]. RAAS activation in heart failure is measured by increased levels of plasma renin and angiotensin, which subsequently leads to vasoconstriction, sodium, and volume retention. Other hormones, including atrial natriuretic peptide, arginine vasopressin, and endothelin-1, are also part of the hormonal activation cascade [8]. Sympathetic activation is measured by increased norepinephrine levels in the plasma, NE spillover, and increased central sympathetic outflow. The NE levels have been shown to be higher in patients with asymptomatic heart failure than in normal controls, and higher in symptomatic heart failure patients than in asymptomatic ones [9]. However, the regional spillover of NE has been shown to have more prognostic value than the systemic levels [6,10]. Most of the regional NE spillover is contributed by the sympathetic nerve terminals in the cardiac and renal tissues [8].

There is a significant interplay between the renal sympathetic activity and the central nervous system creating a continuous feedback loop between the two organs (Figure 1) [11]. Reduced effective plasma volume and cardiac output activate the efferent nerves from the brain, leading to sodium retention and reduced renal blood flow, which triggers the afferent renal nerves to provide feedback to the brain to increase sympathetic activation [7,11,12].

Hormonal and sympathetic activation contribute to myocardial remodeling, which can initially be adaptive in maintaining cardiac output and tissue perfusion. This remodeling involves myocardial hypertrophy and collagen synthesis under the effect of transcriptional protein activation [13]. An excessive and chronic activation can lead to fibrosis, cardiac dilation, and subsequently progression of clinical heart failure. Additionally, the direct cardiotoxic effects of chronically elevated NE may lead to hyper-contracture, and result in irreversible myocardial damage [8,14]. The chemo- and the baroreceptors constitute the peripheral regulators of the cardiac and renal function via the maintenance of vascular tone, and hence organ perfusion is under the influence of the sympathetic and parasympathetic systems [15]. The vasculature is also subject to dynamic changes under the effect of chronic sympathetic and hormonal activation, leading to endothelial cell dysfunction, smooth muscle cell hypertrophy, and vasoconstriction [4]. Chronically elevated plasma NE levels lead to the downregulation of the beta-adrenergic receptors in the cardiac muscle. The regional spillover of NE in the heart also leads to the stores being depleted, hence affecting the uptake in the failing cardiac muscle [7,16].

## 3. Renal Denervation

NE spillover in the kidneys is associated with a worse prognosis [10]. Since the kidneys constitute an integral part of the sympathetic activation feedback loop between the brain, the heart, the vasculature, and the kidneys, the therapies targeting this afferent loop may provide a novel and promising target in the management of patients with heart failure. Renal denervation has evolved from surgical denervation to transcatheter-based technology. This latter technique is a minimally invasive, catheter-based procedure that ablates both the afferent and efferent renal sympathetic nerves [4]. Various catheter-based technologies exist today, including radiofrequency ablation, ultrasound thermal ablation, and chemical ablation utilizing alcohol. Renal denervation has been predominantly studied as a tool for the management of resistant hypertension. While earlier studies failed to demonstrate a clear clinical benefit, with the redefinition of ablation technology and endovascular techniques, more recent trials demonstrated RDN to be able to provide an effective blood pressure control [17,18]. However, multiple concerns remain with RDN use as a therapy for resistant HTN. There has been some heterogeneity in trial design, with occasionally conflicting results. Most of the trials lack long-term follow-up, and not all the trials reported ambulatory blood-pressure measurements. Hence, the effect of RDN on long-term HTN control is unclear. Yet, a recent analysis of nine trials demonstrated a sustained reduction in blood pressure of up to 3 years in some of the trials [19]. The overall complication rate remains low at 1.5%, yet the true burden of long-term complications, such as renal artery stenosis, cannot be fully estimated due to the lack of systematic follow-up imaging of renal arteries in most of the trials [20].

Recently, there has been an increasing interest in RDN as a treatment for chronic heart failure and tachy-arrhythmias, such as atrial fibrillation, with multiple studies demonstrating its efficacy, up to 1-year follow-up [21,22]. As previously described, heart failure syndrome is characterized by an upregulation of sympathetic tone, which, although initially is adaptive, has significant negative pathologic implications long term. Hence, downregulation of the pathological sympathetic signaling may represent a salient therapeutic strategy [23]. To date, there have been no large, randomized, sham-controlled clinical trials evaluating the role of RDN in heart failure patients. However, there have been a number of animal and clinical proof of concept studies which can lay the ground for future randomized trials, to try to validate the role for RDN as an effective heart failure therapy [23,24,25,26,27].

### 3.1. Renal Denervation in Heart Failure with Reduced Ejection Fraction

The ACC/AHA define heart failure with reduced ejection fraction (HFrEF) as systolic dysfunction with left ventricular ejection fraction (LVEF) <40%, with or without symptoms of congestion [28]. As previously discussed, patients with heart failure have a substantial increase in renal norepinephrine spillover, significantly higher than the patients with essential hypertension, due to increased SNS activity. Pharmacotherapies targeting sympathetic over activation have been a cornerstone of heart failure management. Indeed, the current guideline-directed management of HFrEF, with proven morbidity and mortality benefits, revolves around neurohormonal modulators, including beta-blockers, ACEi, ARBs, aldosterone antagonists, diuretics, and neprilysin inhibition [4,28]. However, pharmacotherapy has limitations, including a low patient adherence rate, inability to tolerate the therapeutic doses of medications, polypharmacy with various drug–drug interactions, and side effects. Poor medication adherence (average of 40–60%) is associated with worse heart failure outcomes [29,30]. Hence, non-pharmacological-based therapies targeting these neurohormonal pathways, including RDN, may play an important complementary role in heart failure management. Proof of concept studies were predominantly conducted in animal models of heart failure, including rodent and swine models. Initially, Zheng et al. produced rat heart failure models by inducing MI in Sprague–Dawley rats by coronary ligation [31]. Four weeks later, RDN was performed. The authors demonstrated that RDN lowered norepinephrine and brain natriuretic peptide levels in the hearts of rats with HF. The RDN also decreased the left ventricular end diastolic pressure (LVEDP) and blunted the loss of β1-adrenoceptor β2-adrenoceptor protein expression. The authors concluded that RDN can potentially improve cardiac function mediated by adrenergic agonists and the blunting of β-adrenoceptor expression loss [31]. Polhemus et al. evaluated the cardio-protective effect of RDN on ischemia-reperfusion injury in a rodent model [24]. The hypertensive rats received either bilateral radio frequency (RF) RDN or sham-RDN. At 4 weeks after RF-RDN, the spontaneously hypertensive rats were subjected to 30 min of transient coronary artery occlusion, followed by reperfusion. The authors demonstrated a significant reduction in myocardial infarct size, and preservation of cardiac function 7 days post reperfusion following RDN pretreatment, as compared to a sham control, with reduction in oxidative stress and increased nitric oxide bioavailability [24]. The same group later studied the effects of RDN on LV function and remodeling in a similar ischemia-reperfusion injury rat model [25]. Kyoto rats underwent a myocardial ischemia reperfusion protocol. After 4 weeks, the rats were randomized to sham or radiofrequency RDN. The rats treated with RDN therapy demonstrated a significant improvement in the left ventricular function, vascular reactivity, and reduced cardiac fibrosis [25]. Subsequently, a swine heart failure model was studied to reproduce the rodent findings. Sharp et al. investigated the therapeutic benefits of RF-RDN in normotensive Yucatan swine who underwent the myocardial ischemia reperfusion protocol. The swine with LVEF < 40% underwent blinded randomization to receive sham RDN or RF-RDN treatment, with 12 weeks of follow-up. The authors demonstrated that the RF-RDN therapy resulted in significant reductions in the renal norepinephrine content and circulating angiotensin I and II and an increase in natriuretic peptide levels [23]. Additionally, the LV end-systolic volume was significantly reduced, leading to a marked and sustained improvement in the LV ejection fraction. Additionally, RF-RDN reduced LV fibrosis and improved coronary artery responses to vasodilators. The authors concluded that RDN, via the inhibition of the renal sympathetic activity, leads to attenuation of the renin-angiotensin system activation and improved coronary artery vasorelaxation [23]. Wang et al. redemonstrated similar benefits of RDN in improving heart failure hemodynamics in a dog-heart failure model [32].

It is important to note that, while most of the prior studies demonstrated a favorable RDN effect via the downregulation of SNS, hence the modulation of renal NE, adrenoceptor expression, and RAAS pathway, a novel finding was the effect of RDN on neprilysin levels. Neprilysin is a metallopeptidase responsible for the degradation of several peptides, including ANP, BNP, bradykinin, and Angiotensin 2. Recently neprilysin inhibition, with a concomitant increase in the circulating plasma natriuretic peptides, has become of great interest in the management of HFrEF. Large scale clinical trials, such as PARADIGM-HF, demonstrated that combining a neprilysin inhibitor (Sacubitril) with valsartan was superior to the standard guideline-directed therapy with a RAAS inhibitor alone (enalapril) [33]. Polhemus et al. demonstrated in their rat model that RF-RDN leads to reduced renal nerprilysin levels and increased levels of circulating natriuretic peptides [25]. Later, Sharp et al. demonstrated in swine model that not only did RF-RDN reduce the renal neprilysin levels and increase natriuretic peptide levels, it did so without increasing the angiotensin 2 levels. In fact, the angiotensin 2 levels are decreased after RF-RDN [23]. This combination of downregulation of RAAS and neprilysin pathways potentially have very favorable hemodynamic effects in heart failure.

Apart from the animal models of heart failure, there has been limited clinical data with RDN as a therapy for heart failure (Table 1). Davis et al. reported the first small cohort of seven patients, where RDN was used specifically as a therapy for systolic heart failure [34]. The group demonstrated that RDN was feasible in this small cohort and was associated with no procedural or post-procedural complications. The patient follow up demonstrated improvements in both symptoms and exercise capacity at 6 months [34]. Another human study (PRESERVE) was aimed at evaluating the effect of RDN on renal sodium excretion in HFrEF patients, but was terminated after the negative results from the Symplicity HTN-3 study were published. There have been six other studies which evaluated the outcomes of RDN in patients with systolic dysfunction. Three of these were small-scale randomized studies. Chen et al. randomized 60 HFrEF patients to RDN with medical therapy vs. medical therapy alone (30 patients in each group) [35]. The groups demonstrated that RDN was feasible and safe in this patient cohort and was associated with significant improvement in the left ventricular ejection fraction (LVEF), NT-ProBNP, and LV end diastolic dimensions [35]. Subsequently, Gao et al. reproduced those findings in another small-scale randomized trial of 60 patients (30 in each group). They demonstrated that patients in the RDN group showed a decrease in NT- ProBNP, an increase in LVEF, an improved New York Heart Association (NYHA) class, without reported episodes of hypotension or syncope [36]. A meta-analysis of all of the seven human trials involving RDN use in HFrEF was recently published [26]. The authors demonstrated that bilateral RDN appears safe and well-tolerated in patients with HF and improved the signs and symptoms of HF, slightly decreasing the systolic and diastolic BP without affecting renal function. The pooled-mean NYHA class was significantly decreased, the mean 6-min walk test was increased, and the average NT-proBNP level was decreased. Left ventricular end systolic and end diastolic dimensions were also significantly reduced [26]. It is important to note that the average systolic blood pressure reduction in the RDN group in the analysis was nine mmHg. This latter finding may limit the use of RDN in those patients whose blood pressure is low or low normal at baseline or those who cannot tolerate pharmacotherapy due to hypotension. These small-scale studies with a short-term follow-up period provide hypothesis-generating data that should encourage further larger scale clinical trials with a longer follow-up period to better evaluate the role of RDN in HFrEF management, as an addition to the current guideline-directed pharmacotherapy.

### 3.2. Renal denervation in Heart Failure with Preserved Ejection Fraction

Heart failure with preserved ejection fraction (HFpEF) is a systemic disorder characterized by normal left ventricular ejection fraction, elevated filling pressures, and increased ventricular and arterial stiffness [41]. It is highly prevalent, but generally an under-diagnosed condition with significant morbidity and mortality, and accounts for around half of all heart failure patients. The patients with HFpEF typically have a high prevalence of other conditions, such as hypertension, obesity, sleep apnea, diabetes, chronic kidney disease, and metabolic syndrome [41]. It is important to note the strong association between arterial hypertension, vascular stiffness, left ventricular hypertrophy, and increased sympathetic tone in patients with HFpEF [27]. This increase in ventricular afterload can lead to increased ventricular contractility, myocardial demand, and cardiac remodeling. Unlike HFrEF, the therapies targeting neurohormonal pathways have been less beneficial in HFpEF, most likely secondary to the heterogeneous nature of the disease with various phenotypic manifestations. The pharmacotherapies targeting beta adrenergic receptors, RAAS, or neprilysin pathways have not yielded any major improvement in the outcomes in randomized controlled trials in patients with HFpEF [42,43,44]. It has been noted that the sympathetic nervous system mediates hypertension-induced hypertrophy via the direct stimulation of cardiomyocyte beta-adrenergic receptors. On the other hand, cardiac fibrosis and inflammation is a more heterogeneous process, involving mast cell activation, stimulation of the afferent sympathetic nerves, RAAS activation, and norepinephrine release [45]. Hence, targeting of the afferent signals from the kidney may reduce the sympathetic input to the heart, hence may prove to have a favorable effect on the hemodynamic profile, and prevent cardiac remodeling in those patients. The major evidence for the use of renal denervation in patients with HFpEF comes from the studies of renal denervation used for the treatment of resistant hypertension, due to the high prevalence of HFpEF in the patients enrolled in those studies (Table 2). Previously, Brandt et al. studied a cohort of 46 patients who underwent RDN for resistant hypertension. The group demonstrated that RDN significantly reduced the mean interventricular septum thickness, LV mass index, and was associated with a significant reduction in LV filling pressures and shortening of the isolvolumic relaxation time [46]. The diastolic parameters improved at 1 months and 6 months, post RDN. Importantly, the left ventricular hypertrophy regression was independent of the hypertension control [46]. In a smaller cohort, Mahfoud et al. investigated the effect of RDN on anatomic and functional myocardial parameters, assessed by cardiac magnetic resonance. Out of 55 patients who underwent RDN, 16 met the criteria for HFpEF [47]. The authors showed that there was a significant improvement in the global longitudinal strain, and a reduction in the left ventricular mass index, suggesting improved diastolic function in these patients [47]. More recently, Kresoja et al. examined the effects of RDN in patients with HFpEF. The group retrospectively analyzed patients who underwent RDN and found that approximately 60% of those patients (99 patients) met the criteria for HFpEF. The group demonstrated that the patients with HFpEF undergoing RDN showed increased stroke volume index, vascular, and LV stiffness as compared to the no-HF patients [27]. A reduction in sympathetic nerve activity, ventricular afterload, and arterial stiffness contributed to the restoration of the ventriculo–arterial coupling [48]. Those patients showed symptomatic improvement, and partly reversed the hemodynamic alterations to a level comparable to patients without HF, following RDN. The authors concluded that RDN might be a viable therapeutic strategy for arterial hypertension and HFpEF. Importantly, the observed improvements in those with HFpEF appeared to be independent of the changes in blood pressure. A randomized clinical trial (UNLOAD-HFpEF) is currently being conducted and aims to explore the potential of RDN as a therapy for HFpEF in a single center, pilot trial using a randomized, sham-controlled double-blind design.

### 3.3. Renal Denervation and Cardiac Dysrhythmias in Heart Failure

The patients with chronic heart failure are predisposed to various types of atrial or ventricular tachyarrhythmias, including atrial fibrillation (afib) and ventricular tachycardia. In the patients who suffer myocardial infarction or develop heart failure, autonomic dysregulation leads to an excessive sympathetic drive, and compensatory mechanisms, as previously discussed, become a nidus for cardiac pathology [49]. This sympathetic overstimulation and cardiac remodeling is one of the main drivers of tachyarrhythmias in heart failure. There has been increasing interest in using RDN for the treatment and prevention of arrhythmias and arrhythmia-related morbidity, both atrial and ventricular [50]. The support for this concept was initially demonstrated in animal models, although not necessarily that of heart failure. In a canine model of tachycardia-mediated cardiomyopathy, ventricular remodeling was attenuated by RDN in dogs that were chronically paced at elevated heart rates [51]. In a pig model of myocardial infarction, RDN significantly reduced the occurrence of ventricular fibrillation during ischemia induction, as compared to animals in the sham control group [52]. Similarly, Zhang et al. demonstrated, in a canine model, that surgical and chemical renal denervation decreased whole-body and local tissue sympathetic activity and reversed neural remodeling in the heart and stellate ganglion. RDN was associated with the beneficial remodeling of the infarction zone, translating to a decrease in ventricular arrhythmia after myocardial infarction [53]. Although limited, there are small scale human studies on the effects of RDN on cardiac tachyarrhythmias. Pokushalov et al. randomized a small cohort of 27 patients with afib and hypertension to undergo afib ablation alone, or afib ablation with RDN. The group demonstrated that significantly more patients who underwent concomitant RDN were free of atrial fibrillation at 12 months, compared to those who did not (69% vs. 29%, *p* = 0.033) [54]. ERADICATE-AF was a larger randomized control trial of 302 patients which demonstrated that, among patients with paroxysmal atrial fibrillation and hypertension, renal denervation added to catheter ablation, compared with catheter ablation alone, significantly increased the likelihood of freedom from atrial fibrillation at 12 months (hazard ratio, 0.57; 95% CI, 0.38 to 0.85; *p*  = 0 .006) [22]. More than 75% of the randomized patients had HFpEF with NYHA class 2 symptoms. Hence, the findings of the trials may not apply to lone atrial fibrillation patients. Remo et al. presented four patients with refractory ventricular tachyarrhythmias, despite antiarrhythmic therapy and prior VT ablations (two with ischemic and two with non-ischemic cardiomyopathy). The number of VT episodes decreased from 11.0 ± 4.2 (5.0–14.0) during the month before ablation to 0.3 ± 0.1 (0.2–0.4) per month after ablation. The responses to RDN were similar for ischemic and non-ischemic patients [55]. More recently, in a pooled analysis of 121 patients, Howson and colleagues demonstrated a significant effect of RDN in reducing implantable cardiac defibrillator therapies, reducing the number of VA episodes, antitachycardia pacing and defibrillator shocks [56]. The aforementioned studies represent heterogeneous groups of patients. There is an important need for randomized controlled trials specifically evaluating the antiarrhythmic role of RDN in heart failure patients, with both preserved and reduced EF, for both prevention and as a therapy.

### 3.4. Future Directives and Challenges

Heart failure, regardless of LV function, continues to carry a poor prognosis and is associated with a significant burden on health care systems. Pharmacotherapy targeting various neuro-hormonal pathways was shown to improve clinical outcomes in patients with reduced ejection fraction, yet not so much in preserved LV function. Recently, SGLT2 antagonists have emerged on the front line of pharmacotherapy for heart failure and have shown for the first time an improvement in clinical outcomes in patients with preserved ejection fraction [57,58,59]. However, the recurrent hospitalizations, readmissions, and poor medication compliance due to polypharmacy remain a major limitation in managing these patients. There are several challenges that RDN currently faces in heart failure patients. Heart failure is a very heterogeneous disease and probably only some phenotypes would respond to RDN as a therapeutic modality. Identifying those phenotypes may be challenging [48]. The exact mechanisms through which RDN may exhibits its therapeutic effects in different heart failure phenotypes also remain unclear. While the effects of RDN on hypertension control with afterload reduction may account for some of those mechanisms in HFpEF and select HFrEF phenotypes, the RDN modulation of neurohormonal and cardio-metabolic pathways may play a role in other phenotypes. Our current understanding of the renal nerve anatomy is probably limited. Global renal nerve ablation vs. selective ablation based on nerve fiber composition may be different, hence RDN can produce heterogeneous, afferent sympathetic effects and not offer the same therapeutic benefits in all patients [60]. There may be some variation in the ablation techniques and ablation technology. Finally, defining clinical outcomes in RDN trials may be challenging, with most of the current available evidence on RDN use in heart failure focused on echocardiographic parameter improvement and change in biomarkers, such as natriuretic peptides. Linking RDN to improvement in hard outcomes in heart failure, such as a reduction in hospitalizations or improvements in mortality, will require large randomized trials with sham controls. The associated decrease in BP after RDN may necessitate a reduction in therapeutic doses of heart failure medications, which may also obscure the potential benefits of RDN. Currently, there are two randomized controlled trials with sham control which are ongoing (Table 3). UNLOAD-HFpEF is evaluating RDN in 68 patients with HFpEF, and RE-ADAPT-HF is evaluating RDN use in 144 patients with HFrEF. Both of the trials are sham controlled, and the latter aims at evaluating clinical outcomes including death, hospitalizations, and NYHA class change.

## 4. Conclusions

Chronic heart failure continues to impose a major challenge for clinicians, due to its heterogeneity and the complexity of the syndrome, with incompletely understood pathophysiological mechanisms. The resultant economic burden of the syndrome continues to rise worldwide, as does the prevalence of the disease. The kidneys play a vital role in SNS activation and modulation and are a critical part of the feedback loop between the heart, the brain, and the vasculature. Targeting renal afferent and efferent sympathetic nerves via alternative therapeutic strategies, such as RDN, has proven to be safe, feasible, and efficacious. Through its simultaneous action on various neuro-hormonal pathways, including renal NE spillover, RAAS and neprilysin pathways, RDN has shown to improve ejection fraction, improve diastolic parameters, and regress hypertrophic and maladaptive pathogenic remodeling of the cardiac tissue in animal models and small clinical studies. Additionally, its anti-arrhythmic potential may prove vital in improving heart failure outcomes. RDN offers a therapeutic promise in heart failure patients. We look forward to the results of several larger, randomized controlled trials that are underway to provide further clarity and evidence of the role of RDN in the management of heart failure.

## Figures and Tables

**Figure 1 jcm-11-04147-f001:**
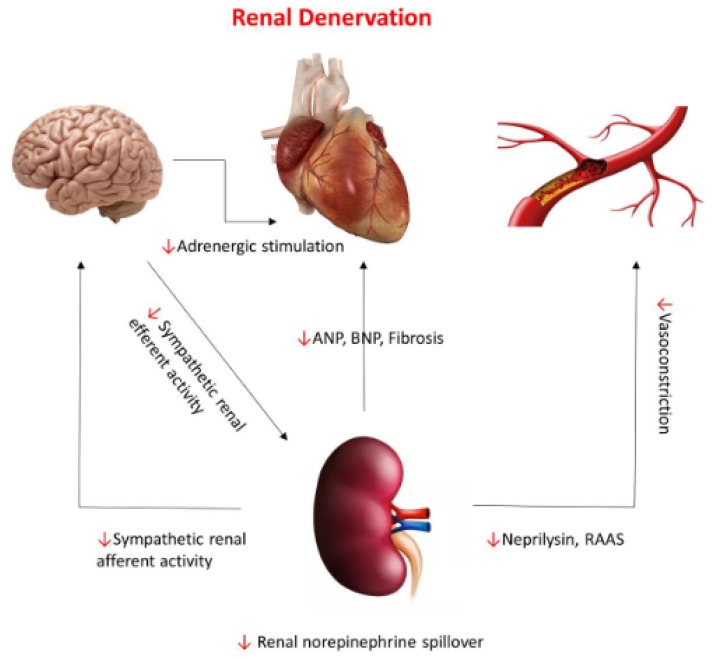
Neuro–hormonal pathways connecting the renal system to the central nervous system, cardiac, and vascular systems and the effect of renal denervation on those pathways (red arrows).

**Table 1 jcm-11-04147-t001:** Use of Renal Denervation in Patients with Heart Failure with Reduced Ejection Fraction.

Study	N, Population	Clinical Findings
Gao et al. (2019) [36]	60, Single-center RCT, EF < 40%	30 patients in RDN group. At 6 months, compared to control group, RDN was associated with significant increase in LVEF, decrease in NT-ProBNP, BP, and NYHA class. No significant changes in glomerular filtration between two groups.
Drożdż et al. (2019) [37]	20, Single-center RCT, EF < 35%	There were no significant differences in LVEF, BP, 6MWT and NT-proBNP concentration at 6 and 12 months after RD or control.
Chen et al. (2017) [35]	60, Single-center RCT, EF < 40%	30 patients RDN group. At 6 months, compared to control group, RDN was associated with significant improvement in symptoms, BP, quality of life, LVEF, NT-ProBNP, and NYHA class. No significant changes in glomerular filtration nor complication of renal artery stenosis were observed.
Gao et al. (2017) [38]	14, Single-arm, EF < 45%	There was a significant decrease in symptoms and improvement in 6-min walk test with increase in LVEF at 6 months follow up. No RDN-related complications were observed during the follow-up period. Additionally, there was significant improvement in BP and GFR remained stable.
Hopper et al. (2017) [39]	39, Multi-center, Single-arm, EF < 40%	RDN was associated with reductions in NT-proBNP and 120-min glucose tolerance test in HF patients 12 months after RDN treatment. No significant change in LVEF, 6 min walk test of GFR.
Dai et al. (2015) [40]	20, Single-center, Single-arm, EF < 40%	No obvious change in heart rate or blood pressure was recorded. Symptoms of heart failure were improved in patients after RDN. No complications were recorded in the study.
Davis et al. (2013) [34]	7, Single-center, Single-arm, EF < 40%	Over 6 months there was a non-significant trend in blood pressure reduction. No hypotensive or syncopal episodes were reported. Renal function remained stable. There was a significant improvement in symptoms and a 6-min walk test.
Xia et al. (2022) [26]	220, meta-analysis of the above studies	Bilateral RDN increased the LVEF, decreased the LVESD, and decreased the LVEDD. In addition, RDN significantly decreased systolic and diastolic BP and decreased HR. RDN did not significantly change GFR or serum creatinine levels. The mean 6-min walk test was increased and NT-pro BNP was decreased.

**Table 2 jcm-11-04147-t002:** Use of Renal Denervation in Patients with Heart Failure with Preserved Ejection Fraction.

Study	N, Population	Clinical Findings
Brandt et al. (2012) [46]	64, Single-center non-randomized study, EF > 55%	46 patients and 18 controls. RDN significantly reduced BP, and LV mass and improved diastolic function at 1 and 6 months.
Mahfoud et al. (2014) [47]	16, Multi-center non-randomized study, EF > 55%	Significant improvement in global longitudinal strain at 6 months. Reduction in left ventricular mass index suggesting an improved diastolic function.
Kresoja et al. (2021) [27]	66, Single center, single-arm, EF > 55%	Patients with HFpEF undergoing RDN showed reduced BP, and increased stroke volume index. LV diastolic stiffness and LV filling pressures as well as NT-proBNP decreased.

**Table 3 jcm-11-04147-t003:** Ongoing Randomized Clinical Trials Evaluating the Use of Renal Denervation in Patients with Heart Failure with Preserved or Reduced Ejection Fraction.

Study	N, Population	Clinical Outcomes
UNLOAD-HFpEF	68, randomized, sham-controlled double-blind design, EF > 55%	Assess the hemodynamic effects of RDN in patients with HFpEF. Effect of RDN on a combination of death, increase in diuretic therapy, hospitalization for heart failure, worsening NYHA-class, change in pulmonary pressure parameters.
RE-ADAPT-HF	144, Prospective, randomized, double-blind, sham-controlled, multicenter. EF < 45%	6-min walk test, Change in NT-pro-BNP, e-GFR, KCCQ at 6 months.

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
