# Peer review of "The Potential Role of Renal Denervation in the Management of Heart Failure"

_jcm, 2022, doi:10.3390/jcm11144147_

Round 1

Reviewer 1 Report

The topic of the manuscript is interesting.

However, there is a too sharp jump to conclusion and to clinical implication of experimental data or evidence in different context (hypertension)

The general tone should be toned down.

More specifi comments:

- page 4 : the picture is far most complex. even if the impact of renal denervation would be considered persistent, with no "escape" over time which implication on BP control? which implication on BP variability? which is the most reliable way to neasure BP inthis context (PMID: 31790375 )?  What about the risk of hypotension - a common phenomenon (PMID: 22227524), that in heart failure adn with advancing age is an equivalent of organ hypoperfusion?

- page 5 lines 82-83. if so, what about the use of  beta-blockers in HF to prevent cardiac remodelling? what should be expected if patient underwent renal denervation?

- page 5 lines 95-99: there are several conflicting results concerning the  issue of safety, efficacy, and persistence of beneftis from renal denervation in HT patietns . Authors discuss only possible strenghts, whereas conflicting and/or negative results deserve more consideration

- page 5 lines 100-  : it is a too simplistic approach. Does atrial fibrillation in HF patients recognize similar pathophysiology thab in patients without HF? ...and what about the issue of the "riming" of atrial fibrilaltiona ppearance in the natural history of HF?

Similarly, a too simplistic approach is adopted at page 11 concerning dysrhythmias

Author Response

The authors would like to thank the reviewer for the very valuable feedback. Please see attached the point-to-point response to each of the points raised. The response appears in red and the corresponding change in the manuscript as highlighted red. 

Reviewer 2 Report

Interesting paper.

the authors show the potential role of renal denervation in the management of the HF.

I like the introduction and the pathophysiology finding.

I agree with the authors on the division of the paragraphs, however I would like to suggest to report in the Future directives and challenges paragraph the ongoing or only designed trials (please in order to increase the interest of readers I would like to add a table with ongoing trials or proposed studies).

Minor comment: please  be more clear, rewrite the definition of HFrEF (reduced EF is systolic dysfunction)

Author Response

The authors would like to thank the reviewer for the very valuable feedback. Please see attached a point-by-point response to each of the comments. The response appears in red and the corresponding change in the text appears in highlighted red. 

Round 2

Reviewer 1 Report

Many concerns have been addressed , largely in a satisfactory manner

I continue to disagree on comments related to page 4. It has not to do with hypertension at all.

If RDN impacts on BP levels and most drugs (starting with diuretics) or interventions used to treat HF impacts on BP levels, then the entire therapeutic strategy and regimen (dosage, etc) is affected and impacted

the way Author replied to that point is not approprriate in my view

Author Response

The authors would like to thank the reviewer for the feedback and comments. 

The authors agree with the reviewer that HTN is an important mediator through which RDN therapy may potentiate its effects in heart failure. And indeed, most of the patients enrolled in the studies presented in the manuscript were hypertensive. Based on the studies presented, the authors want to clarify that heart failure is a very heterogeneous syndrome and in select HF phenotypes, RDN may potentiate its effect predominantly through neurohormonal modulation. As an example, some patients with HFrEF do not have hypertension and are normotensive yet still benefit from drugs like beta blockers and ACEI. On the other hand, in most patients with HFpEF or hypertensive cardiomyopathy patients, HTN maybe the predominant factor through which RDN mediates its therapeutic effects.  We agree with the reviewer that there are nuances regarding the interaction of renal denervation and the medications used to treat heart failure, especially in HFrEF. However, given the limited nature of the current data, we have put in a disclaimer discussing the limited data set regarding adjustments of heart failure medication in the setting of renal denervation as an “adjunctive” treatment in patients with heart failure. 

We have added a paragraph to the future directions and challenges reflecting those points.